# Digital Innovation in Language Teaching—Analysis of the Digital Competence of Teachers according to the DigCompEdu Framework

**María Rubio-Gragera [1], Julio Cabero-Almenara [2] and Antonio Palacios-Rodríguez [2,*]**

[1] Department of Translation and Interpretation, University of Malaga, Av. de Cervantes, 2, 29016 Malaga, Spain; mrubiogr@uma.es

[2] Department of Didactics and Educational Organization, Faculty of Education Sciences, University of Seville, C. San Fernando, 4, 41004 Sevilla, Spain; cabero@us.es

* Correspondence: aprodriguez@us.es

**Abstract:** The utilization of technology in the process of teaching and learning, as well as the influence of the COVID-19 crisis on education, are widely recognized and interconnected factors. This investigation is primarily focused on a group of formal education teachers who have received little attention to date, teachers from Official Language Schools, which are a part of the Special Regime Education system of Andalucía, a southern region of Spain, which provides foreign language education. Specifically, we aim to assess their level of digital proficiency in relation to their experience and use of information and communication technology (ICT) in the classroom. We also analyze how the March 2020 lockdown impacted their confidence levels in utilizing ICT in their teaching practices. One hundred and four teachers took part in the study and answered the DigCompEdu check-in questionnaire. The findings indicate that teachers' overall self-assessment of their digital competence is low, with particular attention needed in the least developed areas, which is the facilitation of digital proficiency to students. Additionally, there are noteworthy differences in the variables of ICT experience and confidence. For example, the amount of time spent utilizing ICT in teaching does not necessarily correlate with teaching proficiency. Based on these results, we discuss potential strategies for enhancing digital competence in this educational group and propose some curriculum content for teacher training in digital competence.

**Keywords:** digital competence of teachers; DigCompEdu; foreign language teaching

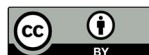

## 1. Introduction

Until March, 2020, the digitalization of the teaching–learning processes could mean innovation within education. However, since the population lockdown caused by the COVID-19 pandemic, the use of digital technology became an essential to implementing educational professional activity and, therefore, the promotion of digital teaching competence (DTC) was a priority objective at all levels and educational contexts. This includes the Official Language Schools, publicly owned centers dedicated exclusively to the teaching of languages, and dependent at the administrative level on the various ministries of education of the different Spanish regions (autonomous communities), framed within what is known as the Special Regime Education system (E.R.E. by its abbreviation in Spanish).

In this sense, digital teaching competence is understood as that set of knowledge, skills and/or abilities that help the teacher to solve pedagogical problems with the help of digital technology [1,2].

This article presents the results of a study carried out to evaluate the level of DTC in the teaching staff of these formal education centers in the autonomous community of Andalusia (south Spain). For this, the following variables have been considered:

1.  The relationship between the DTC level and the seniority/experience of the teachers who implement ICT in class ;
2.  The relationship between the DTC level and the daily amount of time using technology in teaching; and finally,
3.  Teachers' level of confidence using ICT for teaching before, during and after the confinement of the population in March 2020 caused by the COVID19 pandemic.

There are few studies focused on the educational context concerned in this article. Thus, this research aims to contribute to filling the existing gap in the academic research regarding the context of the Official Languages Schools. Particularly, focusing on their digitalization process and, specifically, the digital competences of the teachers of Official Languages Schools.

Nevertheless, studies that support the implementation of educational technology in foreign language teaching do exist, since this plays a relevant role in the domain of language education, and they offer a wide range of possibilities in the classroom environment [1]. Additionally, technology is an effective tool and a significant part of the students' learning process. In this sense, motivation and cooperation are two essential factors in learning language skills, which can be increased through technology [2,3].

Authors such as Benítez and Enríquez [4], for example, defend the use of computer tools both in the preparation of the language lesson and during the lesson itself, in order to make it more effective, enjoyable and interesting. We agree entirely with these authors when they state that not using computer tools would be ignoring an existing reality for students and, therefore, necessary in all spheres of education. Benítez and Enríquez [4] specify two phases in which these tools can be used for the successful teaching–learning process of foreign languages, and which seem to us crucial while designing any teaching proposal. These are the following:

1.  During the preparation of the lesson, look for materials and dictionaries to figure out doubts and questions about the language. We would also add the preparation of the content itself (material, presentations, gamification of activities, etc.).
2.  During the lesson, take advantage of resources such as the Wi-Fi network or the digital devices of the centers, if any are available, as well as the students' own devices, such as cell phones, if possible. In this regard, our view is that, as teachers, we must be cautious and pay special attention to the digital divide between us and the students, as well as the diversity that we can find among students. However, we believe that by designing educational proposals that pay attention to this diversity, the maximum performance could be obtained for the optimization of the teaching–learning process using learning and knowledge technologies (LKT).

Yundayani et al. [5] believe that the use of ICT allows the learning process to be more creative and interactive, putting in practice the four basic aspects of language learning: oral and written comprehension and production. Some of the examples proposed by these authors for learning English as a second language are, specifically, different digital media such as animations, texts, videos, audios, or the combination of all of them (known as multimedia content). Thus, students will acquire, through the implementation of technology, not only purely linguistic skills, but also other skills, such as digital literacy, creativity, reasoning or skills for effective co-communication. These authors list the four stages that, according to Simonson, teachers should pay special attention to while implementing ICTs, as well as in the provision of online materials, during the process of teaching a language:

- Verifying that the technology used is adequate for its aim;

- Determining learning outcomes or, in other words, which are the competencies to be achieved;

- Identifying the learning experiences and customizing them with the available technologies that are intended to be used. Therefore, we must also pay attention to the inclusiveness and accessibility of the tools to ensure that learners receive a meaningful learning experience that has a positive impact in their linguistic skill acquisition process;

- Finally, preparing the learning experiences that are to be presented to students with the main aim of increasing their motivation.

In Andalucía, the digital educational transformation process is a project whose main aim is the digitalization of education in all areas. Even if this is a solvent project, Cabero et al. [6] also identify some weaknesses and threats, such as a needed change in the teaching culture or the provision of material resources to educational centers, which include the ones that are the focus here, the Official Languages Schools.

Additionally, one of the main needs for the smooth functioning of ICT in education is the mastery of educators in digital tools [7].

For this reason, we believe that they the study of the variables observed within this research is relevant in order to know in depth the factors that affect the teaching of digital competences in this particular educational context.

Regarding the relationship between the level of digital competence and their experience implementing ICT for teaching (Variable 1), similar studies to this one carried out with university professors in the field of health sciences affirm that teachers with 4 to 14 years of experience have a better DTC than their more junior and senior colleagues [8]. Other authors, such as Facó Boudet [9], have proved in similar studies that such factors as experience do not influence the level of digital teaching competence and that its development is related to a personal component and a professional engagement. Another similar study is the one carried out by Palau et al. [10] in an educational context similar to ours, the music conservatories, which also belong to the Special Regime Education system. According to this study, less experienced educators consider themselves more competent regarding the implementation of digital technologies.

In the case of the second variable that has been studied, the daily amount of time using technology in the teaching profession, there are also similar studies as the one carried out by Barragán-Sánchez et al. [11], which show that the level of digital competence does not correspond with the amount of time using technology in the classroom.

sierra.sun@mdpi.comdditionally, regarding the impact of the pandemic on the digitalization of the teaching–learning processes (Variable 3), although it is still early to draw general conclusions, some authors, such as Cabero et al. [12], do affirm that compared to the initial moments of uncertainty, bewilderment, anxiety and stress for both teachers and students, the pandemic has had positive repercussions in terms of acquisition of digital skills by teachers, and that there has been a transformation of their initial attitudes regarding the use of technologies for education.

It is assumed that during the lockdown, the digital competences of teachers were crucial in enabling more than 1.37 billion of learners to continue their studies worldwide [13].

As mentioned by Martínez-Garcés and Garcés-Fuenmayor [14] in a study carried out in Colombia, online learning has become a mandatory modality after the repercussions caused by COVID19; therefore, the development and strengthening of the digital competences of educators are essential, considering, according to these authors, that the basic digital competences for academic development are informatization and information literacy, communication and collaboration, digital content creation, digital security and problem-solving skills.

Undoubtedly, as Cabero et al. point out [12], the lockdown should also be a reason to learn to make changes and transformations in the educational system. At the same time, this makes analyzing and investigating the experiences that can be useful in the study of possibly transferring to face-to-face or blended teaching contexts. For all these reasons,

according to these authors, teacher training must become more significant for the incorporation of technology into the teaching–learning processes.

The main objective of this research, in which this paper is framed, is to analyze the digital teaching–learning process in the educational contexts of the Official Languages Schools in the region of Andalucía. Specifically, the digitalization of this process and, therefore, paying special attention to the impact that the pandemic may have had in terms of its forced acceleration. To this end, the specific objective of this research is to analyze the digital competence of a sample group of Official Schools' language teachers through two types of study. On the one hand, a descriptive study based on the DigCompEdu reference model and, on the other hand, through a contrastive study in which the significant differences in the level of digital competence of the sample are studied in terms of the variables experience implementing ICT for teaching purposes, the level of daily use of technology within lessons and the confidence in the implementation of ICT in the classroom before, during and after the confinement caused by the pandemic.

## 2. Materials and Methods

### 2.1. Methodology

Following the criteria established by Hernández, Fernández y Baptista [15], our methodology can be considered quantitative, descriptive and correlational.

### 2.2. Participants

A total of 104 teachers (sample group for convenience) participated in our research. Among them, there were representatives from the 52 Official Languages Schools existing in Andalucía. This group represents 13.28% of the total population, since, according to the data collected from the official website of the Regional Government, the total number of Official Language School teachers for the 2021/2022 academic year was 783.

### 2.3. Instrument

The DigCompEdu check-in questionnaire, developed by Cabero-Almenara y Palacios-Rodríguez was used as an instrument for data collection [3]. This questionnaire, according to its authors, allows educators to better understand the DigCompEdu European framework and helps them identify their level of digital competence, as well as to self-assess their strengths and identify areas of improvement.

The instrument is made up of 67 questions, whose answers are anonymous. Twenty-two of those questions belong to the different competency areas, composing DigCompEdu. These areas do not include only those ones aimed to improve the teaching–learning process specifically, but also those ones that allow teachers to use digital media for their professional interactions in general. According to Cabero-Almenara and Palacios-Rodríguez [6], these areas are the following: professional engagement, digital resources, digital teaching and learning, assessment and feedback, empowering learners, and facilitating learners' digital competence.

The remaining 45 questions are focused on sociodemographic aspects that allow us to study, from a contrastive point of view, how different variables have an impact on the digital competence of the participants.

The instrument was disseminated among the sample group in March 2022 by corporate e-mail. Firstly, through the management teams of each of the 52 centers involved and, secondly, by contacting the different language departments of each center. Likewise, campaigns were carried out to disseminate the questionnaire, as well as the objective of the research and its value for the participants, through social media platforms such as LinkedIn, Facebook or Whatsapp.

### 2.4. Procedure and Data Analysis

Firstly, the reliability and validity data of the questionnaire are provided. For this purpose, a reliability analysis is carried out using Cronbach's alpha and McDonald's omega coefficients for the whole instrument, as well as for the dimensions that it is composed of. After this, to calculate the level of validity, a confirmatory factor analysis was applied using structural equations. In addition, composite reliability (CR), average variance extracted (AVE) and maximum shared variance (MSV) were calculated.

Regarding the objectives, a descriptive analysis of the statistics of the central tendency (mean) and dispersion (standard deviation) was performed to achieve the first specific objective. Finally, to achieve the second specific objective, a nonparametric contrast analysis was performed by calculating the Kruskal–Wallis H test (measuring the experience of using ICT in the classroom and the time of daily use of technology in the teaching profession variables) and the Friedman test (measuring confidence on the use of ICT variable), both with average range analysis. At the same time, it has been verified that the data are not normally distributed through a descriptive study in which skewness and kurtosis have been taken into account. The Kolmogorov–Smirnov goodness-of-fit test confirmed this finding, with significance ($p$-value) equal to 0.000 for all items (non-normal distribution).

## 3. Results

### 3.1. Reliability

The reliability of the questionnaire was calculated by means of Cronbach's alpha and McDonald's omega coefficients, both globally and for each of their dimensions. The obtained results are shown below (Table 1).

**Table 1.** Reliability statistics.

|  | Cronbach's Alpha | McDonald's Omega |
|---|---|---|
| Professional engagement (D_A) | 0.801 | 0.820 |
| Digital resources (D_B) | 0.876 | 0.818 |
| Digital teaching and learning (D_C) | 0.898 | 0.816 |
| Assessment and feedback (D_D) | 0.829 | 0.823 |
| Empowering learners (D_E) | 0.839 | 0.839 |
| Facilitating learners' digital competence (D_F) | 0.901 | 0.912 |
| Total | 0.979 | 0.987 |

According to O'Dwyer and Bernauer [16], values greater than 0.7 indicate high levels of reliability for the questionnaire, both in terms of the instrument as a whole and in relation to the different subsections that it is composed of.

### 3.2. Validity

Table 2 shows the obtained and reference values for the model fitting according to Lévy Mangin et al. [17]: chi-square (CMIN), goodness-of-fit index (GFI), parsimonic goodness-of-fit index (PGFI), normalized fit index (NFI) and normalized par-simonic fit index (PNFI).

**Table 2.** Adjustment index.

| Index | Result | Adjustment |
|---|---|---|
| CMIN | 382.128 | CMIN < 500 |
| GFI | 0.979 | GFI > 0.7 |
| PGFI | 0.786 | PGFI > 0.7 |
| NFI | 0.926 | NFI > 0.7 |
| PNFI | 0.806 | PNFI > 0.7 |

The composite reliability (CR), average variance extracted (AVE) and maximum shared variance (MSV) coefficients are calculated together. Table 3 shows the results, as well as the reference values taken for model adjustment [18].

**Table 3.** Convergent and discriminant validity of the model.

| Area | CR | Adjust | AVE | Adjust | MSV | Adjust |
|---|---|---|---|---|---|---|
| Professional engagement | 0.789 | | 0.642 | | 0.552 | |
| Digital resources | 0.746 | | 0.673 | | 0.523 | |
| Teaching and learning | 0.864 | | 0.69 | | 0.562 | |
| Assessment | 0.858 | CR > 0.7 | 0.684 | CR > 0.5 | 0.418 | MSV < AVE |
| Empowering learners | 0.775 | | 0.643 | | 0.564 | |
| Facilitating learner's digital competence | 0.859 | | 0.679 | | 0.405 | |

All the figures obtained are in agreement with the reference values. Therefore, the reliability of the model (CR), as well as its convergent (AVE) and discriminant (MSV) validity demonstrated are shown.

### 3.3. Description of the Level of Digital Competence

As shown below, the analysis of central tendency (mean) and dispersion (standard deviation) was performed. Table 4 shows the mean values and standard deviations achieved by the participating teachers in the 22 items that make up the DigCompEdu questionnaire.

For a correct interpretation of these data, it is important to consider that the response interval ranges from 0 to 4, which means a total of five response options, 0 being the value given to the lowest level and 4 to the most advanced.

**Table 4.** Descriptive DigCompEdu items.

| Item | M | SD |
|---|---|---|
| A1. I use different digital channels to improve the communication with learners and colleagues, e.g., email, instant messaging apps such as Whatsapp, blogs, school website. | 2.60 | 0.688 |
| A2. I use digital technologies to work together with colleagues inside and outside my educational organization. | 2.22 | 0.930 |
| A3. I actively develop my digital competence for teaching. | 2.68 | 0.956 |
| A4. I am aware of and participate in online training opportunities, such as online courses, MOOCs, webinars, virtual conferences. | 3.13 | 0.991 |
| B1. I use different internet sites and search strategies and select a range of different digital resources. | 2.59 | 0.805 |
| B2. I create my own digital resources and modify existing ones to adapt them to my needs. | 2.55 | 1.083 |
| B3. I effectively protect sensitive content, e.g., exams, grades, personal data. | 2.21 | 0.978 |
| C1. I carefully consider how, when, and why to use digital technologies in teaching, to ensure that they are used with added value. | 2.52 | 1.039 |
| C2. I follow learners' activities and interactions in the collaborative online environments we use. | 2.77 | 1.002 |
| C3. When learners work in groups, they use digital technologies to help them learn and effectively accomplish course tasks. | 2.17 | 0.965 |
| C4. I use digital technologies to allow students to plan, document and monitor their learning themselves, e.g., quizzes for self-assessment, ePortfolios for documentation and showcasing, online diaries/blogs for reflection. | 2.26 | 0.910 |
| D1. I use digital assessment formats to monitor student progress. | 2.25 | 0.875 |
| D2. I analyze all data (information) available to me to timely identify students who need additional support. "Data" includes Students' engagement, performance, grades, attendance, activities and social interactions in (online) environments. "Students who need additional support" are: students | 2.30 | 0.856 |

| | | |
|---|---|---|
| who are at risk of dropping out or underperforming, students who have learning disorders or specific learning needs, students who lack transversal skills, e.g., social, verbal or study skills. | | |
| D3. I use digital technologies to provide effective feedback. | 2.40 | 0.894 |
| E1. When I create digital assignments for learners, I take into account and address potential practical or technical difficulties, e.g., equal access to digital devices, resource interoperability, conversion problems and lack of digital skills. | 2.97 | 1.105 |
| E2. I use digital technologies to offer students personalized learning opportunities, e.g., I give different students different digital tasks to address individual learning needs, preferences and interests. | 2.10 | 0.97 |
| E3. I use digital technologies for students to actively participate in classes. | 2.49 | 0.709 |
| F1. I teach students how to assess the reliability of online information and to identify misinformation and bias. | 1.58 | 0.978 |
| F2. I set up course tasks which require learners to use digital means to communicate and collaborate with each other or with an outside audience. | 1.84 | 0.867 |
| F3. I set up course tasks which require students to create digital content, e.g., videos, audio, photos, digital presentations, blogs and wikis. | 2.25 | 0.757 |
| F4. I teach students how to behave safely and responsibly online. | 1.49 | 1.030 |
| F5. I encourage students to use digital technologies creatively to solve concrete problems, e.g., to overcome obstacles or challenges emerging in the learning process. | 2.18 | 0.864 |

As can be observed, the following items stand out: the supervision of student activities and interactions in online collaborative environments (2.77), addressing problems such as equal access to digital devices and resources, compatibility problems or low levels of student digital competence (2.97) and participation in online training courses by teachers (3.13). On the other hand, the items located at the lowest levels are related to teaching students how to behave safely and responsibly online (1.49), teaching students how to evaluate the reliability of the information found online and identifying erroneous and/or biased information (1.58) and proposing tasks that require students to use digital media to communicate and collaborate with each other or with an external audience (1.84). In this case, all of them are part of the facilitation dimension of digital competence for students.

The descriptive study is completed with the information in Table 5, which shows the results globally and by dimensions.

**Table 5.** DigCompEdu items descriptive analysis.

| Area | M | SD |
|---|---|---|
| Professional engagement (D_A) | 2.66 | 0.680 |
| Digital resources (D_B) | 2.45 | 0.689 |
| Digital teaching and learning (D_C) | 2.43 | 0.709 |
| Assessment and feedback (D_D) | 2.32 | 0.700 |
| Empowering learners (D_E) | 2.52 | 0.755 |
| Facilitating learners' digital competence (D_F) | 1.87 | 0.702 |
| Total instrument (DTC) | 2.37 | 0.559 |

The mean value reached by the teachers in the instrument as a whole was 2.37 with a standard deviation of 0.56, a value that indicates that they position themselves in a central value and, therefore, the perception they have of their DTC mastery is measured.

In addition, from the lowest to the highest, the results by areas/dimensions are rated as it follows:

Area 6 (D_F): facilitating learners' digital competence (1.87).
Area 4 (D_D): assessment and feedback (2.32).
Area 3 (D_C): digital teaching and learning (2.43).
Area 2 (D_B): digital resources (2.45).

Area 5 (D_E): empowering learners (2.52).
Area 1 (D_A): professional engagement (2.66).

These values allow us to point out that, in general, teachers present average levels in all dimensions, without significantly excelling in any of them, which leads us to conclude that the sample would be in need of training in the aforementioned areas.

*3.4. Contrast of the Studied Sociodemographic Variables*

3.4.1. Experience Implementing ICT in Their Lessons

The Kruskal–Wallis non-parametric H test was performed. The results are shown in Table 6.

**Table 6.** Kruskal–Wallis H test "Experience implementing ICT in their lessons".

| | **Test Statistics** | | | | | | |
| --- | --- | --- | --- | --- | --- | --- | --- |
| | **D_A** | **D_B** | **D_C** | **D_D** | **D_E** | **D_F** | **DTC_Total** |
| Kruskal–Wallis H | 4.624 | 8.444 | 3.170 | 11.591 | 6.874 | 5.289 | 7.585 |
| Asymptotic significance | 0.593 | 0.207 | 0.787 | 0.072 | 0.333 | 0.507 | 0.270 |

The contrast test is not significant (sig. > 0.05), neither for the dimensions, nor for the total test. Consequently, it can be stated that there are no statistically significant differences according to teaching experience.

3.4.2. Daily use Time of Technology in Teaching Profession

The Kruskal–Wallis non-parametric H test was performed. The results are shown in Table 7.

**Table 7.** Kruskal–Wallis H test "Daily use of ICT in the classroom".

| | **Test Statisctis** | | | | | | |
| --- | --- | --- | --- | --- | --- | --- | --- |
| | **D_A** | **D_B** | **D_C** | **D_D** | **D_E** | **D_F** | **DTC_Total** |
| Kruskal–Wallis H | 8.960 | 3.445 | 12.045 | 24.207 | 14.196 | 8.426 | 18.677 |
| Asymptotic significance | 0.011 | 0.179 | 0.002 | 0.000 | 0.001 | 0.015 | 0.000 |

The contrast test is significant (sig. < 0.05) for most of the areas (except digital resources), as well as for the total test. Consequently, it can be affirmed that there are statistically significant differences depending on the daily time spent using technology in the classroom. To determine in which group this is more relevant, an average rank analysis was performed (Table 8).

**Table 8.** Average rank analysis "Daily use of ICT in the classroom".

| | **How Much Time Within a Lesson Do You Spend Approximately Using Technology?** | **N** | **Average Range** |
| --- | --- | --- | --- |
| | Generally, I spend some part of the lesson time using technology. | 61 | 55.83 |
| D_A | I rarely spend lesson time using technology. | 22 | 36.45 |
| | I use technology most of the lesson time. | 22 | 61.0 |
| | Total | 105 | |
| | Generally, I spend some part of the lesson time using technology. | 61 | 54.84 |
| D_C | I rarely spend lesson time using technology. | 22 | 35.00 |
| | I use technology most of the time of the lesson. | 22 | 65.89 |
| | Total | 105 | |

| | | | |
|---|---|---|---|
| | Generally, I spend some part of the lesson time using technology. | 61 | 54.98 |
| D_D | I rarely spend lesson time using technology. | 22 | 28.27 |
| | I use technology most of the time of the lesson. | 22 | 72.25 |
| | Total | 105 | |
| | Generally, I spend some part of the lesson time using technology. | 61 | 53.32 |
| D_E | I rarely spend lesson time using technology. | 22 | 35.45 |
| | I use technology most of the time of the lesson. | 22 | 69.66 |
| | Total | 105 | |
| | Generally, I spend some part of the lesson time using technology. | 61 | 55.61 |
| D_F | I rarely spend lesson time using technology. | 22 | 37.02 |
| | I use technology most of the time of the lesson. | 22 | 61.75 |
| | Total | 105 | |
| | Generally, I spend some part of the lesson time using technology. | 61 | 55.54 |
| CDD_TOTAL | I rarely spend lesson time using technology. | 22 | 30.18 |
| | I use technology most of the time of the lesson. | 22 | 68.77 |
| | Total | 105 | |

In all cases, the number of teachers who use technology most of the time during the session is higher than those who use it only some of the time or rarely.

3.4.3. Level of Confidence Implementing ICT for Teaching before/during/after the Lockdown

In order to study whether there are differences in the level of confidence in the use of ICT for teaching before, during and after the lockdown of March, 2020, the Friedman test (nonparametric contrast) was performed. The results are shown in Table 9.

**Table 9.** Friedman test "Level of ICT confidence COVID-19".

| **Test Stadistics** | |
|---|---|
| N | 105 |
| Chi-squared | 101.782 |
| Asymptotic significance | 0.000 |

The contrast test is significant (sig. < 0.05). Consequently, it is affirmed that there are statistically significant differences on the level of confidence in digital technology in teaching before, during and after the lockdown. To specify these results, an average rank analysis is performed (Table 10).

**Table 10.** Average rank analysis "Level of ICT confidence COVID-19".

| **Ranges** | |
|---|---|
| | **Average Range** |
| Level of confidence using ICT for teaching before the March 2020 lockdown. | 1.36 |
| Level of confidence using ICT for teaching during the March 2020 lockdown. | 2.14 |
| Level of confidence using ICT for teaching after the March 2020 lockdown. | 2.50 |

As it can be observed, the highest level of confidence using ICT for teaching is reached after the lockdown (2.50). This fact shows an increase from before the pandemic (1.36) and during the confinement (2.14).

## 4. Discussion

To properly discuss the obtained results from this research, we must refer, On the one hand, to the descriptive study based on the DigCompEdu model and, on the other hand, to the contrastive study in which we study the significant differences in the levels of digital teaching competence of the participants considering their professional experience, their level of daily use of technology and their confidence in the implementation of ICT in their teaching practice before, during and after the lockdown caused by the COVID-19 outbreak.

Firstly, the study that has been carried out has a high level of reliability and validity in terms of the results obtained from the sample. These data allow us to draw comparisons with results obtained in similar studies [8,19–21] to indicate that the DigCompEdu model is quite stable, and to conclude with the statements that are presented below.

On the one hand, the existing need for training trainers, so they can transmit to their students knowledge on the following issues:

1. Cybersecurity;
2. Relevance of documentation processes and identification of fake news;
3. Use of digital media as tools for communication and cooperative work.

As it has been stated above, the perceived lack of the digital competencies of the participating teachers is related to the facilitation of digital competence for students. However, the results show that the general perception that teachers have about their own mastery of digital competence has been measured without significantly highlighting any of the other dimensions studied, which leads us to conclude that there is a need for training the sample in all digital competence aspects.

Regarding the contrastive study. the main conclusions are stated below:

1. There are no significant differences in relation to the variable "experience using ICT in the classroom", which means that the years of experience in the implementation of technology in the classroom does not influence the level of teachers' digital competence;
2. The daily time spent using technology in the classroom does lead to significant differences in terms of the level of digital competence;
3. The March 2020 lockdown represents a turning point in terms of the level of confidence achieved while implementing ICT in the classroom, which was much higher at the times after the COVID-19 crisis.

Another one of the conclusions of our work is related to the fact that the pandemic has served, on the one hand, to increase the digitalization of educational institutions, And, on the other hand, to modify the initial attitudes that teachers had towards technology for teaching–learning purposes. In this sense, this study aims to make visible the obstacles posed by the current educational system. Therefore, it is essential that personalized training plans are worked on, in the same way and, based on the results presented, work must be carried out on the necessary accreditation and recognition of digital teaching competence [22,23].

The main limitation of this study is linked to the difficulty to reach the sample. As mentioned above, the research regarding contexts belonging to the Special Regime Education system is very scarce. However, the obtained sample allows us to draw some general conclusions since it represents all Andalucian regions.

As future research lines or projects, we consider the design of a specific training plan for educators related to DigCompEdu levels, which will be focused on the needs identified in this research. As planned by [24,25], this training plan could be designed in the form of a t-MOOC or similar, so that it would be a relevant tool with the possibility of great reach

within the educational community. Similarly, future lines of research may be linked to the longitudinal study of digital competence [26] and the integration of emerging technologies for language learning [27,28] or content creation [29].

**Author Contributions:.** Conceptualization, M.R.-G. and J.C.-A.; methodology, A.P.-R.; software, A.P.-R.; validation, J.C.-A., M.R.-G. and A.P.-R.; formal analysis, M.R.-G.; investigation, M.R.-G.; resources, J.C.-A.; data curation, A.P.-R.; writing—original draft preparation, M.R.-G.; writing—review and editing, M.R.-G.; visualization, J.C.-A.; supervision, J.C.-A.; project administration, J.C.-A.; funding acquisition, A.P.-R. All authors have read and agreed to the published version of the manuscript.

**Funding:** This research was funded by the Margarita Salas Grants, a program for the requalification of the Spanish university system, NextGeneration EU Funds (European Union), linked to the R+D+I DIPROMOOC project (RTI2018-097214-B-C31).

**Institutional Review Board Statement:** Not applicable.

**Informed Consent Statement:** Informed consent was obtained from all subjects involved in the study.

**Data Availability Statement:** https://bit.ly/3nkEFFK.

**Conflicts of Interest:** The authors declare no conflict of interest.

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
