# Peer review of "Digital Innovation in Language Teaching—Analysis of the Digital Competence of Teachers according to the DigCompEdu Framework"

_education, doi:10.3390/educsci13040336_

Round 1

Reviewer 1 Report

I enjoyed reading your paper. The topic of the article is interesting and a timely study. At first glance, it seems to offer a significant perspective on the study of teacher digital competence, with a great contribution to the educational community.

The document is well-structured, easing the comprehension of the study conducted.

The literature review is relevant and up to date. However, I recommend incorporating some more work on levels of digital competence in teachers (https://doi.org/10.6018/reifop.542191).

The research problem and the objectives of the study are well defined and clearly achievable.

The research phases are presented in a clear and structured way. With respect to the sample, it is sufficient. The data collection instrument (questionnaire) is well documented and validated.

The results are presented in a descriptive way for each dimension of the instrument, which facilitates understanding.

The discussion of the data is dynamic with respect to the results obtained in similar studies.

The conclusions are clearly specified and respond to the four objectives of the study. However, I recommend include in future lines other options for working with technologies and language learning, such as "game-based learning" (https://doi.org/10.24310/innoeduca.2022.v8i2.13858) or augmented reality (https://doi.org/10.24310/innoeduca.2021.v7i1.9823).

Reviewer 2 Report

The article is very interesting and up to date on covid-19 and the digital education process throughout that time.
The paper is structured well, easy to read, brings light to the topic and contributes to alternatives in the educational process.
It follows the right structure of research, but the literature review should be more extensive.
The questionnaire is well documented and validated and the sample is enough satisfied.
 Results and discussion are presented in the right research way compared with similar studies.

Reviewer 3 Report

Thank you for the good paper. I hope that my comments will increase the readership and usefulness of your article. In any case, strengthening the theoretical foundations, better explaining the methodology,  can help to strengthen your work. 

Is the manuscript clear, relevant for the field and presented in a well-structured manner?
The paper contains new and significant information adequate to justify publication. The content of the article has followed a new approach, but it needs improvement in theoretical foundations and methodology. 

Are the cited references mostly recent publications (within the last 5 years) and relevant? Does it include an excessive number of self-citations?
The article should further clarify how digital competences are understood and their importance. It would be worth mentioning that they are now an important element of key competences as well as a part of transversal competences. That is why their education is so important. The article presents an overview of current and relevant articles, but please also note the following texts:
Castañeda, L., & Villar-Onrubia, D. (2023). Beyond functionality: Building critical digital teaching competence among future primary education teachers. Contemporary Educational Technology, 15(1), ep397. https://doi.org/10.30935/cedtech/12599

Skevi, O., Ortega-Martín, J. L., & González-Gijón, G. (2023). Use of ICTs and the Digital Competences of Foreign Language Teachers before and during the State of Alarm. Language Related Research, 14(1), 145–166. https://doi.org/10.29252/LRR.14.1.6

Verdecho, M.-J., Alfaro-Saiz, J.-J., Rodríguez-Rodríguez, R., & Gómez-Gasquet, P. (2021). Using an ANP performance management framework to manage the development of transversal competences in University degrees. Central European Journal of Operations Research, 29(4), 1329–1352. https://doi.org/10.1007/s10100-020-00693-7

Is the manuscript scientifically sound and is the experimental design appropriate to test the hypothesis?
The research model is correct and allows the hypotheses to be verified, but the basis of the study in the field of methodology needs correction. Why to use the method, theoretical bases for using the tools as well as what are the main research questions that need to be corrected.  The authors indicated the objectives of the study, the main goal " is to improve the teaching-learning process in the educational contexts of the Official Languages Schools in the region of Andalucía." Isn't that a practical goal? Therefore, it is worth pointing out recommendations for schools on what and how to improve and how to implement it? In order to provide better suggestions, refer to the analysis of the results and provide suitable practical suggestions from the findings of the research.
Shouldn't the verification of the tool also be a specific objective? Its relevance and reliability? The results section suggests this.

Are the manuscript’s results reproducible based on the details given in the methods section?
The results of the work are repeatable based on the indicated methods.  The presentation of the results must be improved as new research goals and questions emerge. My only recommendation is: 1) this part will also require reconstruction according to the structure of the research objective - research question - result.

Are the figures/tables/images/schemes appropriate? Do they properly show the data? Are they easy to interpret and understand? Is the data interpreted appropriately and consistently throughout the manuscript? Please include details regarding the statistical analysis or data acquired from specific databases.

The results were presented correctly and legibly, and data were analyzed appropriately.The tables are clear and correctly describe the reliability and validity results obtained. Evaluation of the author's reliability and accuracy of the questionnaire was carried out correctly and the obtained results confirm its credibility.

Are the conclusions consistent with the evidence and arguments presented?
Although the paper clearly identifies any implications for research, as I have already mentioned, the author/authors should emphasize the practical/application goal more in the part devoted to the results. It is worth developing more specific recommendations. Studies in these fields need a look that can help different areas such as policy makers and practitioners in the area of education.
